# Winter is coming–Temperature affects immune defenses and susceptibility to *Batrachochytrium salamandrivorans*

**Edward Davis Carter**[1], **Molly C. Bletz**[2], **Mitchell Le Sage**[3], **Brandon LaBumbard**[2], **Louise A. Rollins-Smith**[3], **Douglas C. Woodhams**[2], **Debra L. Miller**[1,4], **Matthew J. Gray**[1]*

**1** Center for Wildlife Health, University of Tennessee Institute of Agriculture, Knoxville, Tennessee, United States of America, **2** Department of Biology, University of Massachusetts Boston, Boston, Massachusetts, United States of America, **3** Departments of Pathology, Microbiology and Immunology and of Pediatrics, Vanderbilt University School of Medicine and Department of Biological Sciences, Vanderbilt University, Nashville, Tennessee, United States of America, **4** Department of Biomedical and Diagnostic Sciences, College of Veterinary Medicine, University of Tennessee, Knoxville, Tennessee, United States of America

* mgray11@utk.edu

**Data Availability Statement:** All data not provided in the manuscript is available https://doi.org/10.7290/t7SaLlfxxe.

## Abstract

Environmental temperature is a key factor driving various biological processes, including immune defenses and host-pathogen interactions. Here, we evaluated the effects of environmental temperature on the pathogenicity of the emerging fungal pathogen, *Batrachochytrium salamandrivorans* (*Bsal*), using controlled laboratory experiments, and measured components of host immune defense to identify regulating mechanisms. We found that adult and juvenile *Notophthalmus viridescens* died faster due to *Bsal* chytridiomycosis at 14˚C than at 6 and 22˚C. Pathogen replication rates, total available proteins on the skin, and microbiome composition likely drove these relationships. Temperature-dependent skin microbiome composition in our laboratory experiments matched seasonal trends in wild *N. viridescens*, adding validity to these results. We also found that hydrophobic peptide production after two months post-exposure to *Bsal* was reduced in infected animals compared to controls, perhaps due to peptide release earlier in infection or impaired granular gland function in diseased animals. Using our temperature-dependent susceptibility results, we performed a geographic analysis that revealed *N. viridescens* populations in the northeastern United States and southeastern Canada are at greatest risk for *Bsal* invasion, which shifted risk north compared to previous assessments. Our results indicate that environmental temperature will play a key role in the epidemiology of *Bsal* and provide evidence that temperature manipulations may be a viable disease management strategy.

## Author summary

In 2010, a new skin-eating fungus, *Batrachochytrium salamandrivorans* (*Bsal*), was discovered killing salamanders in the Netherlands. Since then, the pathogen has spread to other European countries. *Bsal* is believed to be from Asia and is being translocated through the

**Funding:** MJG, DLM, DCW, and LAR-S received funds from the National Science Foundation Division of Environmental Biology (Ecology and Evolution of Infectious Diseases Program, https://www.nsf.gov/funding/pgm_summ.jsp?pims_id=5269) Grant #1814520. DCW and LAR-S were partially supported by the U.S. Department of Defense Strategic Environmental Research and Development Program (https://www.serdp-estcp.org/) Grant #W912HQ-16-C-0033). MJG and DLM were supported by U.S. Department of Agriculture National Institute of Food and Agriculture (https://nifa.usda.gov/program/hatch-act-1887-multistate-research-fund), Hatch Project #1012932. The funders had no role in study design, data collection and analysis, decision to publish, or preparation of the manuscript.

**Competing interests:** The authors have declared that no competing interests exist.

international trade of amphibians. To our knowledge, *Bsal* has not arrived to North America. As a proactive strategy for disease control, we evaluated how a range of environmental temperatures in North America could affect invasion risk of *Bsal* into a widely distributed salamander species, the eastern newt (*Notophthalmus viridescens*). Our results show that northeastern USA, southeastern Canada, and the higher elevations of the Appalachian Mountains have the greatest likelihood of *Bsal* invasion, when temperature-dependent susceptibility is included in risk analyses. Changes in eastern newt susceptibility to *Bsal* infection associated with temperature are likely an interaction between pathogen replication rate and host immune defenses, including changes in skin microbiome composition and the host's ability to produce *Bsal*-killing proteins on the skin. Our study provides new insights into how latitude, elevation and season can impact the epidemiology of *Bsal*, and suggests that strategies that manipulate microclimate of newt habitats could be useful in managing *Bsal* outbreaks and that climate change will impact *Bsal* invasion probability.

## Introduction

The environment and interactions among organisms under local abiotic conditions are key factors underpinning the functional ecology and distribution of organisms [1]. In particular, temperature drives many ecophysiological processes that manifest in organismal vital rates (of both hosts and pathogens), and can influence population persistence and species distributions. Temperature influences organisms through various mechanisms, including metabolic and immunological pathways [2]. It also can drive interactions among organisms and microparasites [3]. Thus, environmental temperature is expected to affect host susceptibility to pathogens and epidemiological processes.

By their nature, the ecophysiology of ectothermic vertebrate species is strongly coupled with environmental temperature [2,4]. For example, environmental temperature affects many physiological processes in amphibians including immune function [5,6]. Temperature also drives the diversity of microbes on amphibian skin, which afford protection against many microparasites [7], and can affect the physiology and growth rate of microparasites [8]. Indeed, temperature is considered a primary factor influencing invasion probability of pathogens in ectothermic vertebrate populations [9,10].

In the last 20 years, two pathogenic skin fungi, *Batrachochytrium dendrobatidis* (*Bd*) and *B. salamandrivorans* (*Bsal*), have emerged globally in amphibian populations [11,12]. These fungi, which cause the disease chytridiomycosis, have recently been described as causing the greatest known loss of vertebrate biodiversity attributed to disease [13]. *Bd* has contributed to the decline of hundreds of frog species, while *Bsal* has caused devastating population impacts in several European salamander species since its initial description in 2013 [12,13,14,15]. New research suggests that *Bsal* has been in Europe since at least 2004 [16]. Scheele *et al.* [17] suggested that environmental variables are key to determining invasion risk of chytrid fungi.

Given the strong coupling between environmental temperature and *Bd* epidemiology [18], it is likely that temperature will influence *Bsal* disease dynamics. *Bsal* growth in culture occurs between 5–25°C [12,19]; however, it is expected that successful infection *in vivo* is more restrictive due to interactions with host immune defenses [20]. Previous studies with the European fire salamander (*Salamandra salamandra*) suggest temperature can affect the pathogenicity of *Bsal* [15,19,21], although possible mechanisms have not been identified. Several studies suggest that skin microbial communities play a large role in defending host species from

cutaneous fungal pathogens, such as *Bd* [22,23], and temperature can affect the composition and antifungal function of microbes found on amphibian skin [22,24–26]. Antimicrobial peptides, host-derived enzymes such as lysozymes, and antimicrobial products of symbiotic organisms also play an important role in host immune defense against *Bd*, and their functionality may be driven by temperature [5,27]. Understanding the impact of environmental temperature on the temporal and spatial niche of *Bsal* [28], and the immunological mechanisms that drive these relationships is fundamental to identifying geographic locations where pathogen invasion is most likely and to conceiving disease management strategies.

We undertook a multi-scale, transdisciplinary disease investigation to quantify the effects of three biologically relevant environmental temperatures (6, 14 and 22˚C) on host susceptibility to *Bsal* infection and the development of chytridiomycosis. Given previous work on *in vitro* growth of *Bsal* and heat treatment of infected hosts [12,15,19–21], we hypothesized that *Bsal* would be most pathogenic at 14˚C and least pathogenic at 22˚C. We expected intermediate results at 6˚C, because *Bsal* replication occurs, albeit at a lower rate than at 14˚C. Given the temperature dependent nature of amphibian immune function [5,27], we hypothesized that differences in host susceptibility would be driven in part by reduced innate immune response at colder exposure temperatures. Lastly, we hypothesized that we would observe temperature-dependent patterns in the skin microbiome as observed by others [7], possibly due to microbial thermal optima. We used our estimates of temperature-dependent susceptibility to map the invasion probability of *Bsal* into North America for a widely distributed host species, the eastern newt (*Notophthalmus viridescens*). We investigated temperature-dependent trends in susceptibility for adult and juvenile (eft) life stages for this species. Our results broadly inform possible biological feedbacks among environmental temperature, host immune defenses and susceptibility to pathogens, and will help with planning disease mitigation for the emerging fungal pathogen, *Bsal*.

## Results

### Survival and infection load

Dose-dependent mortality was observed at 6 and 14˚C for both adult and eft *N. viridescens*, with 100% mortality at the two highest exposure doses (Fig 1 and S1 Table). No mortality was observed for adult *N. viridescens* exposed at 22˚C; however, 67% of efts experienced mortality at this temperature at the highest zoospore dose ($5x10^6$). Survival duration for adult *N. viridescens* differed among temperatures for all zoospore doses, with the shortest survival duration

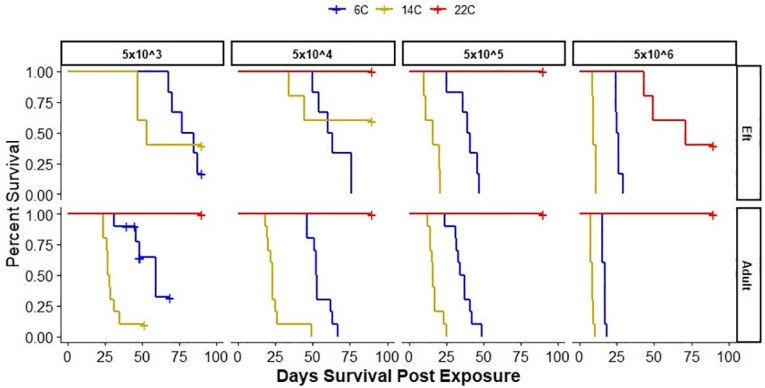

**Fig 1. Survival curves representing the mortality rate of *Notophthalmus viridescens* held at 6, 14, and 22˚C and exposed to four *Batrachochytrium salamandrivorans* doses ($5x10^{3-6}$ zoospores per 10 mL).** The survival curves for adult and eft (juvenile) *N. viridescens* are shown in the lower and upper panels, respectively.

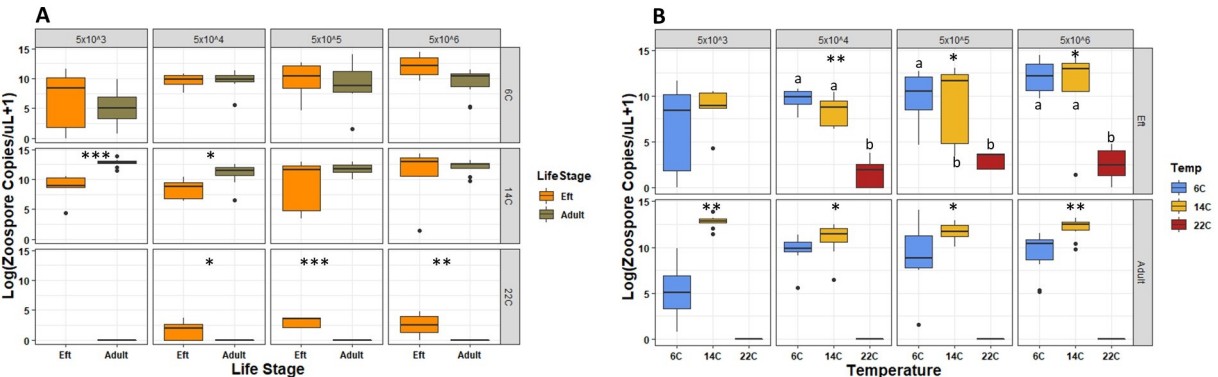

**Fig 2. Boxplots showing log-transformed *Bsal* zoospore copies/uL+1 estimated at necropsy from swab samples tested via qPCR.** Boxplots shown represent each zoospore exposure dose (5x10^{3-6} per 10 mL) for each life stage (**A**) and temperature (**B**). Boxplots faceted by temperature and exposure dose are colored brown and orange in (**A**) representing adult and eft life stages, respectively. Boxplots faceted by life stage and exposure dose are colored blue, yellow and red in (**B**) representing 6, 14 and 22˚C exposure temperatures, respectively. Significant Wilcoxon tests between exposure temperatures for each exposure dose are shown in (**A**) (ns = not significant, * = $P<0.05$, ** = $P<0.01$, *** = $P<0.001$). Significant Kruskal-Wallis tests are indicated in (**B**) (ns = not significant, * = $P<0.05$, ** = $P<0.01$, *** = $P<0.001$). Temperatures with unlike lowercase letters are significantly different.

experienced by those exposed to the highest zoospore dose. When holding dose constant, adult *N. viridescens* exposed at 14˚C were approximately 11X (95% CI, 5.96, 20.171; $P<0.001$) more likely to die each day compared to those at 6˚C. Across all doses, adult *N. viridescens* lived around 20 days longer at 6˚C compared to those exposed at 14˚C. Survival of efts was also temperature dependent, especially at the two highest zoospore exposure doses. Efts exposed at 14˚C were approximately 3X (95% CI, 1.23, 6.78; $P = 0.015$) and >200X more likely to die each day than those exposed at 6 and 22˚C, respectively. Keeping dose and temperature constant, adults were approximately 4X (95%CI, 2.67,6.75, $P<0.001$) more likely to die than efts (Fig 1 and S1 Table).

For adult *N. viridescens*, *Bsal* zoospore load on their skin at necropsy was greater at 14˚C compared to 6˚C (Fig 2 and S2 Table). None of the adult *N. viridescens* exposed at 22˚C developed detectable infections; however, infection was detected in 81% of efts. *Bsal* loads for efts exposed at 6 and 14˚C were significantly greater than at 22˚C (S2 Table and Fig 2).

We confirmed *Bsal* infection and chytridiomycosis in all *N. viridescens* that died using histopathology (Fig 3). We also confirmed the absence and presence of infection in *N. viridescens* that survived and tested qPCR negative and positive, respectively, for *Bsal* DNA.

## Skin microbiome

Newt bacterial communities were composed predominantly of taxa from the phyla Proteobacteria and Bacteroidetes, and fungal communities were composed of taxa from the phyla Ascomycota, Basidiomycota, Blastocladiomycota, and Basidiobolomycota. Relative abundance of bacterial and fungal orders within these groups varied among temperatures (Fig 4A and 4D). For example, Flavobacterales (bacteria) and Blastocladiales (fungi) were more abundant on adult *N. viridescens* at 22˚C compared to individuals at 6˚C and 14˚C.

Bacterial community richness and phylogenetic diversity of adult *N. viridescens* skin bacterial communities differed among temperatures (Richness: LR $\chi^2$ = 120.3, $P < 0.001$, PD: LR $\chi^2$ = 94.8, $P < 0.001$), with individuals at 14˚C having greater richness and diversity compared to those at 6 and 22˚C (Fig 4B). Fungal richness also varied among temperatures (Richness: LR $\chi^2$ = 7.275, $P = 0.026$), with *N. viridescens* at 22˚C having greater fungal richness compared to those at 14˚C (z-ratio = 2.69, $P = 0.02$; Fig 4E). Skin bacterial and fungal community structure

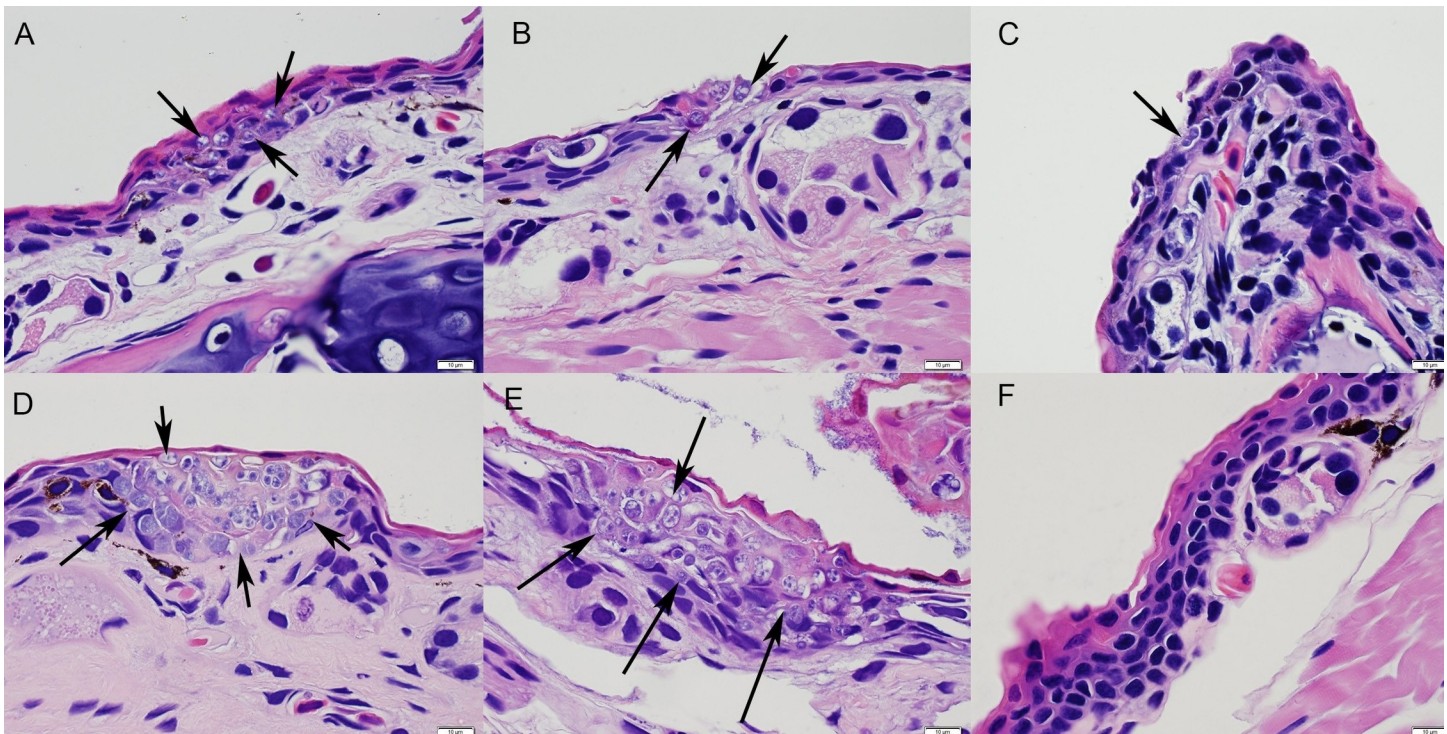

**Fig 3. Hematoxylin and Eosin stained sections of skin from *Notophthalmus viridescens* exposed to *Batrachochytrium salamandrivorans* (*Bsal*) zoospores (doses = 5x10$^{3-6}$ per 10 mL).** Fungal organisms (arrows) were observed invading the epidermis of juveniles (efts) at all temperatures, 6, 14, and 22˚C (**A**, **B** and **C**, respectively), but only at 6 and 14˚C for adults (**D** and **E**, respectively). No invasion was observed in adult skin exposed at 22˚C, even at the highest dose of 5x10$^6$ *Bsal* zoospores (**F**).

differed among temperatures (16S weighted Unifrac—$R^2$ = 0.17, Pseudo-$F$ = 12.8, $P$ < 0.001, 16S unweighted Unifrac—$R^2$ = 0.33, Pseudo-$F$ = 31.7, $P$ < 0.001, Fig 4C and ITS Jaccard: $R^2$ = 0.085, Pseudo-$F$ = 4.031, $P$ < 0.001, Fig 4F). *N. viridescens* at 22˚C exhibited more distinct communities compared to individuals at both 6 and 14˚C. Similarly, the magnitude of difference in fungal community composition was greatest for *N. viridescens* at 22˚C. Group dispersion also differed significantly among temperatures for bacterial (unweighted Unifrac, $F_{124,2}$ = 21.54, $P$ < 0.001) and fungal communities ($F_{86,2}$ = 12.46, $P$ < 0.001). For bacterial communities, dispersion at 14˚C was significantly greater than at 22 and 6˚C ($P$ < 0.001). For fungal communities, dispersion was greater in both 14 and 6˚C compared to those at 22˚C ($P$ <0.01). Predicted *Bsal*-inhibitory function differed significantly among temperatures, with *N. viridescens* at 22˚C having the highest predicted function, followed by 6 and 14˚C (LR $\chi^2$ = 13449.4, $P$ < 0.001, Fig 4B).

The skin bacterial community structure of wild caught *N. viridescens* differed with *Bd* infection status (weighted Unifrac: $R^2$ = 0.03, Pseudo-$F$ = 3.13, $P$ = 0.014; unweighted Unifrac: $R^2$ = 0.03, Pseudo-$F$ = 2.78, $P$ = 0.001, Fig 5A), among seasons ($R^2$ = 0.12, Pseudo-$F$ = 6.78, $P$ = 0.001, $R^2$ = 0.09, Pseudo-$F$ = 4.13, $P$ = 0.001, Fig 5B), and in *Bd*-inhibitory predicted function ($R^2$ = 0.06, Pseudo-$F$ = 7.02, $P$ = 0.001, $R^2$ = 0.02, Pseudo-$F$ = 2.39, $P$ = 0.001). In the wild, *Bd*-infected *N. viridescens* and uninfected *N. viridescens* had significantly different microbiome structure ($R^2$ = 0.07, Pseudo-$F$ = 6.41, $P$ = 0.001), spring was significantly different than summer ($R^2$ = 0.13, Pseudo-$F$ = 12.03, $P$ = 0.003), and uninfected *N. viridescens* had higher *Bd*-inhibitory predicted function than infected individuals (LR $\chi^2$ = 9.51, $P$ = 0.002).

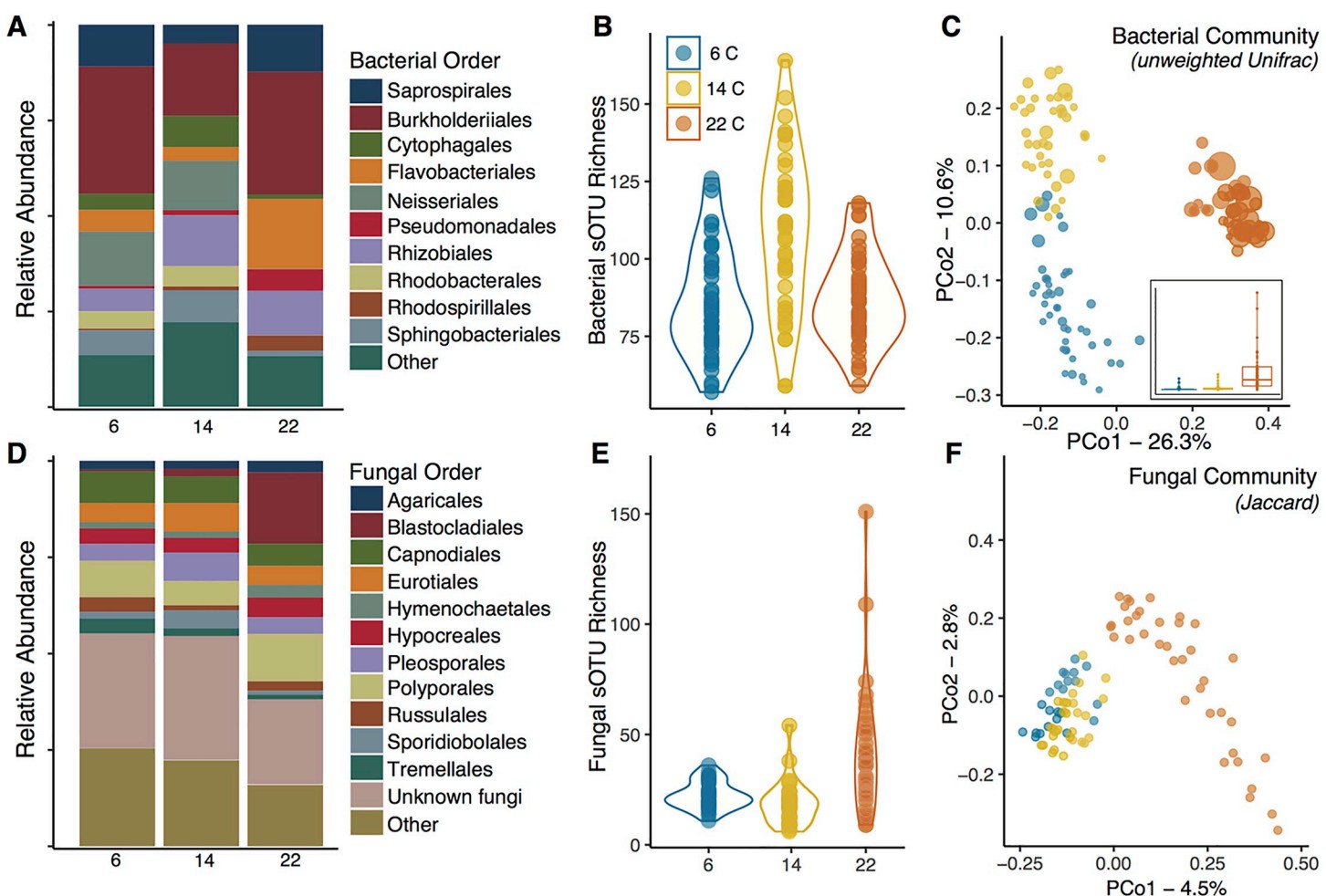

**Fig 4. Skin microbiomes of adult *Notophthalmus viridescens* housed at three experimental temperatures. (A)** Taxonomic composition of bacterial communities at the order level varied across temperatures; **(B)** sOTU richness of skin bacterial communities differed among temperatures; **(C)** Principal coordinates analysis (PCoA) of unweighted Unifrac distances showed differences in bacterial community composition between temperatures. Points are scaled by predicted *Bsal*-inhibitory function. Inset plot shows the same scaling factor (predicted *Bsal*-inhibitory function) across temperatures, with the horizontal lines denoting the median values. **(D)** Taxonomic composition of fungal communities at the order level varied across temperatures; **(E)** sOTU Richness of fungal communities differed among temperatures; **(F)** Principle coordinates analysis of binary Jaccard distances showed differences in fungal community composition between temperatures.

## Skin proteins and peptides

Total skin secretion protein recovery at 6˚C, measured as µg collected corrected by newt body weight in grams (ug/gbw), was greater than at 22˚C for adult control *N. viridescens* (Fig 6A–6C and S3 Table). No differences were detected among temperatures in hydrophobic molecule recovery, percent inhibition, or effectiveness for control *N. viridescens* (S1 Fig). At 6˚C, recovered hydrophobic molecules were greater (ug/gbw) in control *N. viridescens* compared to exposed *N. viridescens*, but no differences were detected in recovered total proteins (Fig 7).

## Temperature adjusted *Bsal* habitat suitability modeling

Invasion risk of *Bsal* into *N. viridescens* populations in North America, adjusting for temperature-dependent susceptibility, indicated that the greatest risk is in the northeastern United States and southeastern Canada, and along the higher elevations of the Appalachian Mountains (Fig 8). These results also suggest there could be seasonal fluctuations in outbreaks,

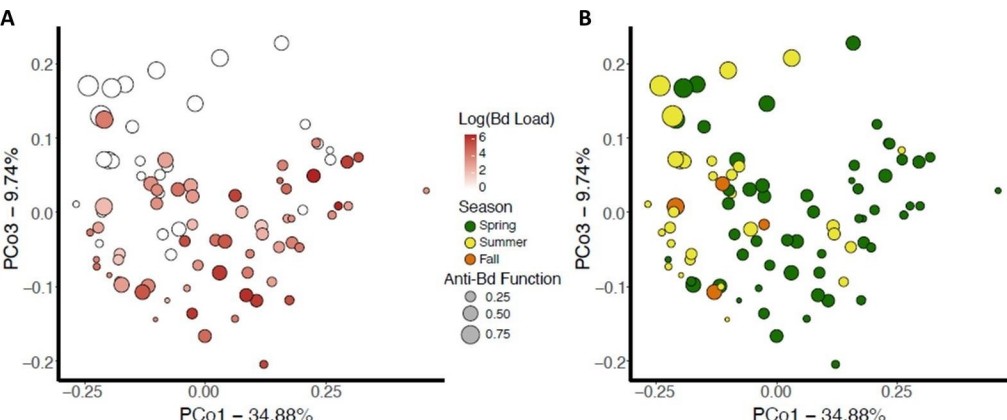

**Fig 5. Skin microbiome community structure from wild adult *Notophthalmus viridescens* ($n$ = 89) was plotted by PCoA of weighted Unifrac distances and examined for patterns with (A) *Bd* infection load (Uninfected = white to $3*10^6$ = red), and (B) seasonal shifts in the microbiome.** Each point represents an individual newt microbiome and is size-scaled based on the relative abundance of predicted *Batrachochytrium*-inhibitory function. Predicted inhibitory function was greatest in the summer and for uninfected *N. viridescens*.

where geographic locations that exceed 22˚C, could be at lower risk (e.g., southeastern United States).

## Discussion

Similar to research with European *S. salamandra* [15,19,21], our results suggest environmental temperature plays a critical role in the epidemiology of *Bsal*, which could manifest into

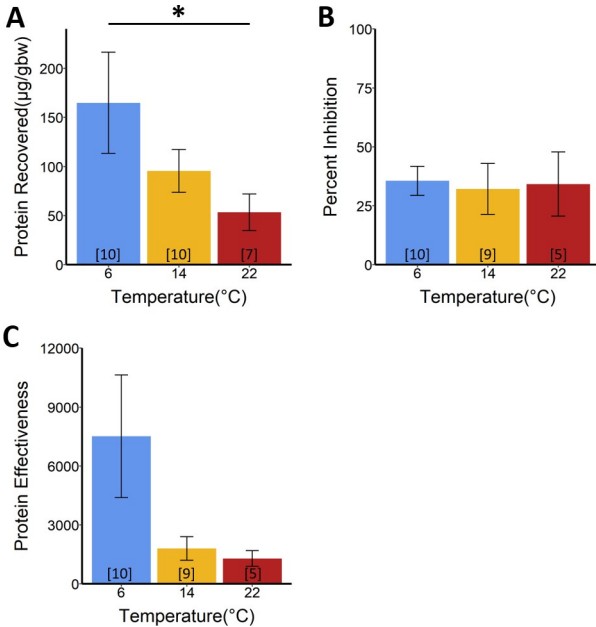

**Fig 6. Skin proteins recovered and inhibition of *Bsal* zoospore viability by total skin secretions from adult *Notophthalmus viridescens* not exposed to *Bsal*.** (A) Total proteins recovered by temperature corrected for body mass (ANOVA; $F_{2,24}$ = 4.08, $P$ = 0.03; see S1 Table for post-hoc analysis), (B) Percent inhibition of *Bsal* zoospore viability by concentrated skin secretions (500 μg/mL) by temperature when challenged in a CellTiter-Glo assay (ANOVA; $F_{2,21}$ = 0.04, $P$ = 0.96), (C) Protein effectiveness (total recovered proteins x percent inhibition of *Bsal*) for three temperatures (Kruskal-Wallis rank-sum test; $\chi^2_2$ = 4.57, $P$ = 0.10). Numbers in brackets indicate sample size.

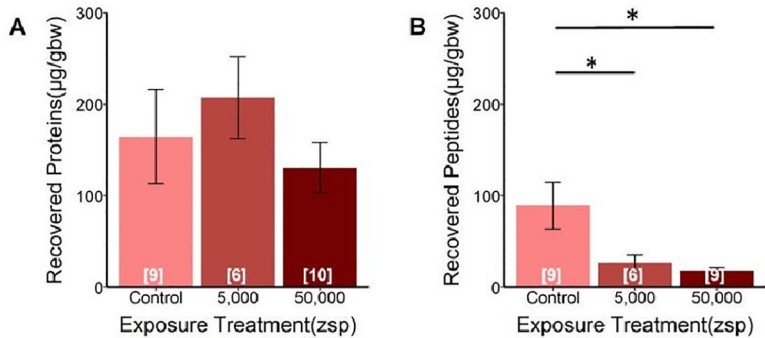

**Fig 7. Concentrations of (A) recovered proteins and (B) recovered hydrophobic molecules from adult *Notophthalmus viridescens* skin secretions corrected for body mass (μg/gbw), with increasing *Bsal* zoospore (zsp) exposure doses in comparison with controls (no exposure).** All *N. viridescens* were held at 6°C. There was no effect of *Bsal* exposure dose on the amount of total proteins recovered (**A**: ANOVA; $F_{2,22}$ = 0.72, $P$ = 0.50). *Bsal* exposure dose had a significant effect on the amount of putative hydrophobic peptides recovered from skin secretions (**B**: ANOVA; $F_{2,21}$ = 5.11, $P$ = 0.02). Tukey Multiple Comparison test indicated that unexposed (control) *N. viridescens* had significantly more peptides recovered than those exposed to *Bsal*. Numbers in brackets indicate sample size.

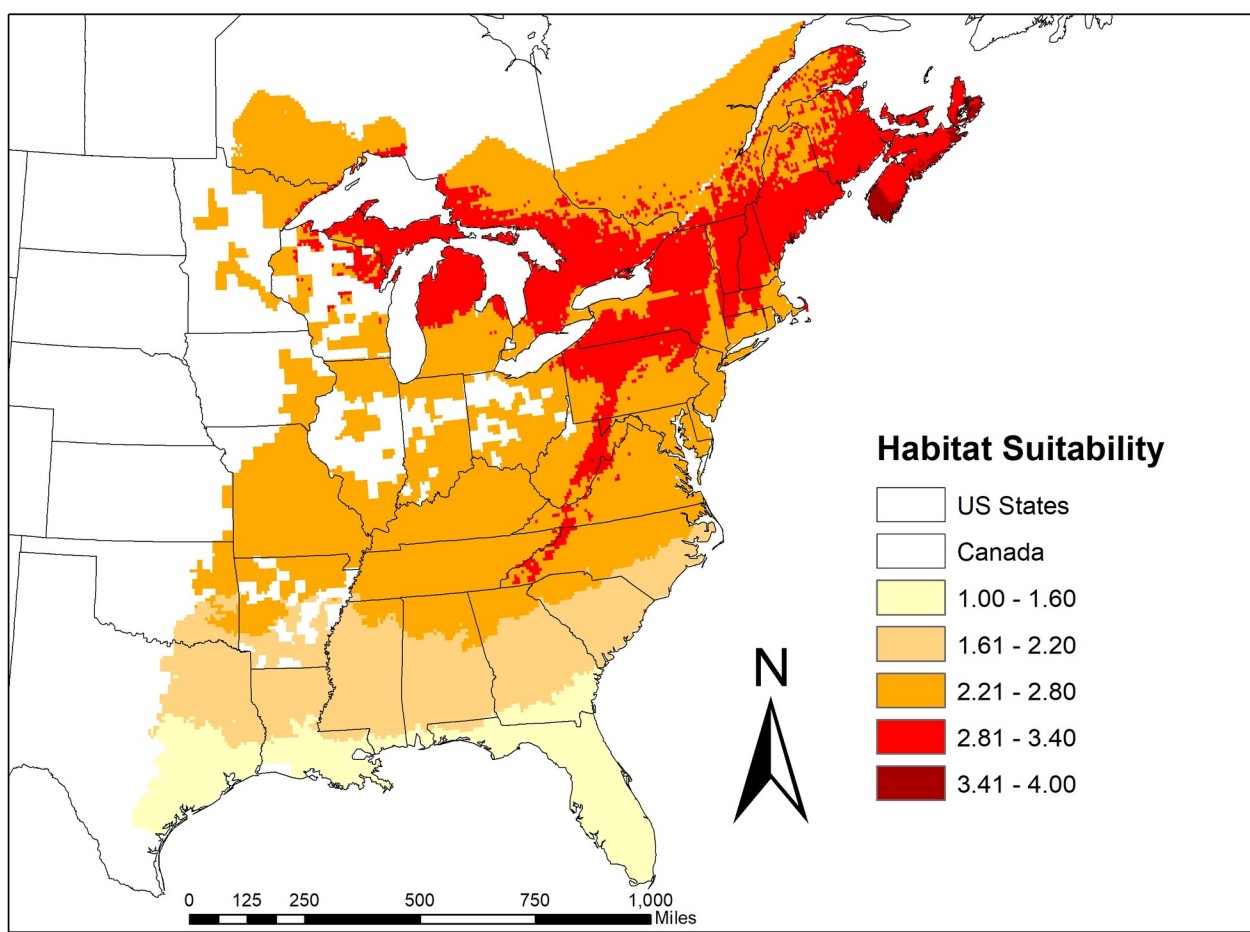

**Fig 8. Invasion risk of *Bsal* into *Notophthalmus viridescens* populations in North America, adjusting for temperature-dependent susceptibility.** Red indicates increasing risk, while light-colored areas indicate low invasion risk. Regions shaded white are outside of the *N. viridescens* range.

geographic variation in outbreaks [19]. *Bsal* invasion into *N. viridescens* populations in North America is expected to be greatest in the northeastern United States and southeastern Canada, and at the higher elevations of the Appalachian Mountains where the species is present. Previous risk analyses suggested that the southeastern United States was at greatest risk of *Bsal* invasion [29,30]; however, these analyses did not consider the impacts of temperature on species susceptibility. Our results suggest that climate change could impact *Bsal* invasion risk and geographic spread [31]. They also suggest that habitat manipulations that increase water temperature (e.g., tree canopy reduction) might be a viable *Bsal* management strategy [32], although the influence of fine-scale temperature manipulations on likelihood of *Bsal* invasion needs to be explored.

Temperature-dependent susceptibility to *Bsal* also could manifest into seasonal fluctuations in outbreaks [19]. Although seasonal trends *Bsal* chytridiomycosis have not been reported yet in Europe, silent declines could be occurring in autumn and winter without detection because many salamander species reduce activity, become fossorial, or use deeper locations of water bodies when temperatures become colder [33]. Indeed, Beukema et al. [19] reported that *Bsal* growth on salamanders in Europe followed temperature-dependent trends. If seasonality plays a role in *Bsal* epidemiology, it is possible that outbreaks in *N. viridescens* populations could decrease during summer months and increase as winter approaches. Further, *N. viridescens* at 6 and 14˚C experienced >90% mortality but died 1.4X slower at the lower temperature. Thus, infected individuals could remain on the landscape for longer duration, increasing opportunity for disease spread and death during winter, analogous to a "White Walker Effect" (*sensu* HBO Game of Thrones). In contrast, we observed limited infection and mortality at 22˚C, suggesting that monitoring for *Bsal* infection and outbreaks should be concentrated in geographic regions where and during months when average ambient temperature is <22˚C.

Several mechanisms may have been responsible for the temperature-dependent trends we observed in *N. viridescens* susceptibility to *Bsal*. Although we did not measure *in vitro* growth, others have demonstrated maximum growth of *Bsal* at 15˚C, and growth decreased by 70% at 22˚C and 50% at 5˚C. Therefore, *Bsal* replication rate could have played a role in both the near lack of disease at 22˚C and reduced mortality rate at 6˚C compared to 14˚C. At 22˚C, Martel et al. [12] also reported that motile zoospores in culture were not observed; reduced zoospore motility is associated with reduced pathogenicity of *Bd* [34]. Indeed, heat treatment of *S. salamandra* for 10 days at 25˚C has been shown to clear *Bsal* infections [20]. Beukema et al. [19] also showed that salamanders exposed to *Bsal* in thermal gradient enclosures and regulated their temperatures between 17.5 and 21.6˚C did not become infected, which they attributed in part to inhibition of *Bsal* growth.

While clearly pathogen replication rates can impact survival of infected hosts [35], temperature-dependent trends in susceptibility also could be related to host immune defenses. We found that microbial community composition on the skin of *N. viridescens* differed among temperatures. Most notably, the relative abundance of bacteria inhibitory to *Bsal* at 22˚C was greater compared to the lower temperatures in both our laboratory experiments and field surveys, suggesting that temperature-mediated shifts in skin microbiome may contribute to host protection. Previous studies have shown the bacterial community on amphibian skin can influence host survival when exposed to fungal pathogens [36–38]. Increased fungal diversity at 22˚C could also contribute to host protection, as many fungi produce inhibitory compounds [39]. Interestingly, we found that total proteins recovered on the skin of *N. viridescens* were lowest at 22˚C and no differences were detected in hydrophobic peptides production among temperatures. Thus, reduced host susceptibility to *Bsal* at higher temperatures is likely an interaction between shifts in microbiome composition and pathogen replication rates rather than antimicrobial skin secretions produced by the host [12,19].

In contrast to our hypothesis, *N. viridescens* were able to produce substantial antimicrobial secretions even at the coldest temperature. Total recovered proteins on *N. viridescens* skin at 6˚C were greater than at 14 and 22˚C, which may have resulted in greater inhibition of *Bsal* zoospores and slowed the rate of disease progression at 6˚C. Proteins produced by amphibians or in the metabolites of resident skin microbes can have inhibitory effects on *Bd* [40,41]. The skin microbiome was similar between 6˚C and 14˚C, hence it was likely not a mechanism influencing differences in mortality rate at these temperatures.

A component of defensive skin secretions in many frog species includes low mass hydrophobic peptides [41]. Much less is known about defensive peptides or proteins in the secretions of salamanders [42–44]. At 6˚C, the amount of defensive hydrophobic molecules produced after two months exposure to *Bsal* was approximately 5X less than in control *N. viridescens*, suggesting that available defensive molecules were released from granular glands during early stages of infection, and supporting the hypothesis of reduced growth of *Bsal* at 6˚C. It also is possible that defensive peptide production became impaired later in infection, because *Bsal* can invade the granular glands and possibly disrupt function [45]. Collectively, these results suggest that the putative hydrophobic peptide component of skin defenses may be most effective during initial phases of *Bsal* infection, but that chronic infections, such as those occurring at 6˚C, can eventually overwhelm salamander host defenses.

Although adult *N. viridescens* died slower at 6˚C than at the other temperatures, they also died at lower pathogen burdens suggesting lower infection tolerance [46]. On average, *N. viridescens* at 6˚C had 4X lower *Bsal* loads on their skin at necropsy compared to those exposed at 14˚C, which is a similar trend with *S. salamandra* [15]. Amphibians subjected to cold temperatures may experience immunosuppression leading to decreased infection tolerance [47–49]. Several *Bd* challenge experiments have demonstrated temperatures below the optimal *in vitro* growth temperature for *Bd* (17–23˚C) can cause high host mortality at lower infection intensities [3,50–52]. These studies suggested that reduced mortality and higher *Bd* infection intensity at higher exposure temperatures were in part due to temperature-dependent infection tolerance of hosts. Thus, even though *Bsal* growth is slower and inhibitory proteins on newt skin are greater at temperatures below 14˚C, their ability to tolerate skin damage may be less as ambient temperature drops, contributing to the White Walker Effect. It is also possible that cooler temperatures create more ideal conditions for secondary bacterial infections through the necrotic ulcerations that *Bsal* creates [53].

Efts followed similar temperature-dependent infection and survival trends, although they on average died approximately 1.4X slower than adults, suggesting that life stages may play a role in *Bsal* epidemiology, as also seen with *Bd* and ranaviruses [18,54]. The mechanisms driving the difference in susceptibility between life stages are unknown (i.e., we did not measure immune defenses in efts), but may be influenced by habitat conditions, tetrodotoxin (TTX) in their skin, or skin microbial composition and function. Adult and eft *N. viridescens* were housed in containers with water and moist paper towel, respectively, similar to their natural aquatic and terrestrial habitats. It is possible that aquatic environments facilitated reinfection and disease progression due to release of flagellated zoospores that transmit efficiently through water [12]. *N. viridescens* efts also have approximately 10X the concentration of TTX in their skin as adults [55]. While TTX does not appear to have direct effects on *Bsal* growth (DCW, unpubl. data), it might interact with the skin microbiome to create more *Bsal*-inhibitory effects [56,57]. Lastly, efts and adults were collected in different U.S. states (Pennsylvania and Tennessee, respectively) due to availability, hence it is possible that immunological co-factors associated with geography played a role in affecting *Bsal* susceptibility.

Collectively, our results demonstrate environmental temperature will be a critical determinant of *Bsal* disease dynamics for *N. viridescens* and possibly other amphibians. Given that *N.*

*viridescens* are distributed across most of eastern North America, invasion risk will likely vary among seasons, latitudes, and elevations similar predictions in Europe [19]. Our results possibly provide support for the chytrid thermal optimum hypothesis [58]; however, more research is needed. Although the temperatures used in our experiments are environmentally relevant for eastern North America, they do not reflect the expected variation that *N. viridescens* could experience in a day or among months. Indeed, future experiments should explore the role of temperature variation on susceptibility to *Bsal* and impacts on host immune response. Further, support for the thermal mismatch hypothesis should be explored [3,59].

## Materials and methods

### Ethics statement

All husbandry and euthanasia procedures followed recommendations provided by the American Veterinary Medical Association and the Association of Zoos and Aquariums. All animal procedures were approved by the University of Tennessee Institutional Animal Care and Use Committee under protocol #2623. *N. viridescens* that reached euthanasia endpoints were humanely euthanized via water bath exposure to benzocaine hydrochloride.

### Animal challenge experiments and analyses

The model species we used was *Notophthalmus viridescens* (eastern newts), which is one of the most widely distributed amphibian species in North America [60], and is susceptible to *Bsal* chytridiomycosis [61,62]. This species has two distinct post-metamorphic life stages: terrestrial juvenile (called an eft) and aquatic adult [63]. Efts can remain terrestrial for up to seven years before returning to aquatic sites to breed as adults [63]. Given this complex life cycle and the significant role both could play in *Bsal* epidemiology, we decided to tested the susceptibility of both life stages. Based on availability, we collected adult and eft *N. viridescens* from aquatic and terrestrial habitats in Tennessee and Pennsylvania, respectively (TN Scientific Collection Permit #1504 and PA Scientific Collection Permit 2019-01-0082). Upon arrival to the laboratory, we heat-treated all *N. viridescens* for 10 days at 30°C to clear any potential preexisting *Bd* infections [64,65], because chytrid coinfections could influence *Bsal* disease outcomes [62]. Prior infection with *Bd* and subsequent clearing, however, does not seem to influence susceptibility to *Bsal* infection in *N. viridescens* [62]. We confirmed that all *N. viridescens* were *Bd* negative by testing pre-trial skin swabs using quantitative polymerase chain reaction (qPCR) methods similar to those described in Boyle et al [66] and following the same genomic DNA extraction methods described below. The pre-trial skin swabs and all swabs collected following *Bsal* exposure consisted of 10 firm ventral passes along the salamander's abdomen followed by 10 swipes on the bottom of each foot. After heat treatment, *N. viridescens* were acclimated for one week to one of three temperatures (6, 14 and 22°C), which corresponded to average ambient temperatures during winter, spring/autumn, and summer, respectively, in mid-latitudinal North America [67].

We followed a standard protocol for host-pathogen challenges using a multidose-response experiment to evaluate how *N. viridescens* susceptibility and *Bsal* pathogenicity were influenced by temperature [68,69]. For the adult experiments, we randomly assigned 10 animals to one of four exposure doses ($5x10^3$, $5x10^4$, $5x10^5$ and $5x10^6$ *Bsal* zoospores per 10 mL) and assigned five animals as controls (*n* = 45 for each temperature trial; S1 Table). We performed each temperature experiment within a Conviron environmental growth chamber (Winnipeg, Canada) set to the target temperature and light/dark cycles reflecting the corresponding season (winter = 8 hours light; spring/autumn = 10 hours; summer = 12 hours light). We housed adult *N. viridescens* individually in sterilized circular 2000-cm$^3$ containers that had 300 mL of

water and a semicircular PVC cover object. The container and water were changed every three days, and we fed the salamanders bloodworms (*Chironomus plumosus*) at 2% of their body mass 24 hours prior to each water change.

Efts were randomly assigned to the same exposure doses used for the adults, but sample sizes were smaller (*n* = 5–6 animals per dose, S1 Table) due to availability. Unlike the aquatic adults, efts live in terrestrial habitats, hence they were housed in sterilized 710-cm³ containers that included a moist paper towel and PVC cover object. We changed each eft's container, paper towel, and PVC cover object every 3 days. All animals were fed 2% of their body mass in fruit flies (*Drosophila melanogaster*) every 3 days.

The *Bsal* we used was isolated (isolate AMFP13/1, [12]) from a morbid *S. salamandra* in the Netherlands. We maintained *Bsal* cultures at the University of Tennessee and enumerated zoo-spores following methods described in Carter et al.[68]. All animals were exposed for 24 hrs to their randomly assigned zoospore dose, and controls were inoculated with autoclaved dechlo-rinated water [68].

The experiments lasted between 45 and 90 days, which has been shown to be a sufficient duration for *Bsal* chytridiomycosis to develop [19,62,68]. We humanely euthanized animals after the loss of righting reflex, which is a terminal endpoint to *Bsal* chytridiomycosis. As such, we recorded individuals that were euthanized prior to the end of an experiment as mortality events for data analyses.

To detect infection and estimate pathogen loads, we swabbed each individual every six days and at necropsy. Quantitative PCR was run on each swab following methods described in Carter et al.[68]. We stained cross-sections of the skin with hematoxylin and eosin stain to confirm *Bsal* infection and chytridiomycosis [70].

We used Kaplan-Meier survival analysis to test for differences in mortality rates among *Bsal* doses for each temperature and between temperatures for each dose. Kaplan-Meier survival analyses were also used to test for differences between the life stages (adults vs. efts) exposed to the same dose at the same temperature. We used Cox-proportional hazard models to robustly estimate the magnitude of difference between life stages, temperatures, and doses that were significantly different. All analyses were performed using the "survival" package in R (version 3.5.3, [71]), and significant differences declared at $\alpha = 0.05$.

To understand the relationship between temperature and pathogen burden on infected hosts, we estimated *Bsal* loads (DNA copies/uL) at necropsy and compared among doses and temperatures using Kruskal-Wallis tests, because the data did not follow a normal distribution [72]. Wilcoxon signed-rank tests that were Bonferroni corrected were used for pairwise comparisons of treatments if the Kruskal-Wallis test was significant [72].

## Microbiome analyses

We quantified differences in skin bacterial and fungal communities among temperatures for adult *N. viridescens* by sequencing extracted gDNA from swabs taken after one-week acclimation to temperatures but prior to *Bsal* exposure. We also sampled 89 adult *N. viridescens* in the wild at Camp Johnson (Colchester, Vermont, USA) and Ethan Allen (Underhill, Vermont) across seasons and evaluated the microbiome in relation to *Bd*-infection (VT Scientific Collection Permit #SR-2016-17; see S1 Text for details). We PCR-amplified the V4 region of the 16S rRNA gene for bacterial communities and the ITS1 region for fungal communities with single index barcoded primers (bacterial: 515f–806r, fungal: ITS1f-ITS2r) as described in the Earth Microbiome Protocol (EMP, 2018) with a few modifications to increase PCR efficiency. The PCR cycle conditions followed Bletz et al. [73] for bacterial community amplification and the EMP for fungal communtiy amplification. PCR products were purified and equalized using

the Omega Biotek MagBead Normalization Kit (Omega biotek, Norcross, GA USA). We pooled equal-molar PCR products and these were sequenced on an Illumina MiSeq (Illumina, San Diego, CA USA) at University of Massachusetts Boston using 2x250 paired-end v2 chemistry kit.

We processed sequence data in Quantitative Insights into Microbial Ecology 2 (QIIME2, Bolyen et al. [74]). Briefly, we demultiplexed, quality filtered (minimum q-score of 10), and clustered raw sequence data into sub-Operational Taxonomic Units (sOTU) using deblur for bacterial communities [75] and DADA2 for fungal communities [76]. For ITS data, we trimmed the conserved flanking regions from sequences using ITSxpress as recommended for amplicon sequencing [77,78]. After filtering, 1,242,115 bacterial 16S sequence reads remained (average: 9,269.5 reads/sample) representing 1,142 sOTUs and 895,217 fungal ITS reads (average 6,781.9 reads/sample) and 1,901 ASVs. We subsequently rarefied all samples at 5,000 reads per sample for bacteria and 1,000 reads for fungal communities to adequately capture sample diversity and retain the majority of sequenced samples. We assigned taxonomy with classify-sklearn [79], built a phylogenetic tree using mafft and fasttree2 [80], and calculated alpha and beta diversity metrics (sOTU richness and Faith's phylogenetic diversity), weighted and unweighted UniFrac metrics (16S only), and Bray-Curtis and binary Jaccard metrics in QIIME2.

Statistical analysis and visualizations were performed in R [81]. We used generalized linear mixed models (GLMMs) to compare alpha diversity across temperatures (including collection site as a random effect; [82]), and completed pairwise post-hoc tests using the "emmeans" package. To compare beta diversity among temperatures, we performed permutational multivariate analysis of variance (PERMANOVA), and performed pairwise post-hoc tests using the pairwiseAdonis() function [83]. We tested for multivariate homogeneity of variance (i.e., group dispersion) using the betadisper() function [84].

We predicted *Bsal*-inhibitory function using a custom-made database containing 16S rRNA sequences from amphibian skin bacteria (including eastern *N. viridescens*) that have been tested for functional activity against *Bsal* [40,53]. We determined which sOTUs in the dataset potentially had *Bsal*-inhibitory function by clustering sOTU data from the Illumina amplicon sequencing at a 99% similarity threshold to the sequences of the strictly inhibitory cultures in the database. Subsequently, we calculated the relative abundance of this function (i.e., proportion of "inhibitory" reads with respect to the full, rarefied community) and richness of *Bsal*-inhibitory sOTUs, and compared these values among temperatures using GLMMS. Similarly, field sampled *N. viridescens* microbiomes were examined for anti-*Bd* function by matching reads to sequences from approximately 7000 cultures tested against *Bd* [40]. Importantly, we acknowledge that the function of microbial secondary metabolites can differ among *Bd* isolates [85], and 16S rRNA sequences do not faithfully indicate secondary metabolite production, though function may be a phylogenetically conserved trait [86]. Thus, ascribing function of microbial communities comes with uncertainty, but may be useful for between-group comparisons [87].

### Defensive proteins and peptide analyses

We quantified and compared total proteins in lyophilized skin secretions and hydrophobic molecules enriched by C18 Sep-Pak passage from the skin of adult *N. viridescens* among the three target temperatures for control and exposed animals that survived to the end of the experiment. These *N. viridescens* were a subset of those described above; however, they were collected from the same site, treated identically (except they were not swabbed), and tested at the same time. To induce the release of skin secretions, we injected acetylcholine (ACh) at 25 nmol per gram body mass using a 1 mL tuberculin syringe with a 25-gauge needle. After

injections, we transferred the *N. viridescens* to a sterile sample bag containing 15 mL of HPLC-grade water and immersed them for 15 minutes to collect secretions. We split the water into two fractions: one which was immediately frozen at -20˚C after taking a small sample for protein quantification, and another that was acidified with concentrated trifluoroacetic acid (TFA, 1% by volume) to inhibit proteases. Protein content of the frozen sample was quantified using the Thermo Scientific microBCA protein-assay kit with Bovine Serum Albumin (BSA) as a protein standard (66.5 kDa). The frozen samples were lyophilized and rehydrated to a concentration of 1 mg/mL protein content in HPLC-grade water. The acidified samples were enriched for hydrophobic molecules using Waters Sep-Pak C18 cartridges (Waters Corporation, Milford, MA, USA), and a small volume of enriched product was removed for quantification. We then dehydrated the samples by vacuum centrifugation. The samples containing possible hydrophobic peptides were quantified by microBCA using bradykinin as a standard (~1kDa) as previously described [41,88]. All hydrophobic samples were reconstituted to 1 mg/mL in HPLC water. All protein and putative peptide samples were stored at -20˚C until tested in zoospore inhibition assays.

To determine the effect of the lyophilized total skin secretions on the viability of *Bsal* zoospores, we used the Promega CellTiter-Glo Luminescent Cell Viability Assay (Promega Corporation, Madison, WI, USA). For each assay, 25 μL of zoospores ($1x10^6$ zoospores/mL) was added to 25 μL of the skin secretions (500 μg/ml final concentration) in triplicate for one hour, and the reactions were treated with the CellTiter-Glo reagent. After 15 minutes of incubation, luminescence was determined using a BioTek Synergy 2 plate reader (Biotek Instruments, Winooski, VT, USA). Percent inhibition of *Bsal* was determined by comparing the luminescence of samples in which *Bsal* zoospores were challenged with skin secretions to the luminescence in an unchallenged live *Bsal* zoospore control.

To determine the effect of the hydrophobic fraction of newt skin secretions on the viability of *Bsal* zoospores, we conducted growth inhibition assays as previously described for *Bd* zoospores with minor changes [89]. Briefly, three replicates of $5 x10^4$ zoospores in 50 μL of TGhL broth (16 g tryptone, 4 g gelatin hydrolysate, 2 g lactose per liter of distilled water) were incubated with 50 μL of enriched hydrophobic skin fraction in TGhL broth at final concentrations of 500, 250, 125, 62.5, 31.25, 15.6, and 7.8 μg/mL for seven days. The assay plates were read for absorbance at 490 nm on a BioTek Synergy 2 plate reader. Percent inhibition of *Bsal* by hydrophobic fraction was determined by comparing the absorbance of samples in which zoospores were challenged with skin secretions to the absorbance in a *Bsal* control growing in broth only.

For control *N. viridescens*, we used analysis-of-variance (ANOVA) to test for differences among temperatures in lyophilized skin secretions and hydrophobic molecules recovered, and in percent inhibition of *Bsal* zoospores. To test for differences in effectiveness of the lyophilized proteins or enriched hydrophobic fractions of skin secretions (i.e., the product of recovery and percent inhibition) among temperatures, we used either ANOVA or a Kruskal-Wallis rank-sum test. For 6˚C *N. viridescens*, we were also able to use ANOVA to test if total lyophilized skin secretions and putative hydrophobic peptides that were recovered differed between control and exposed *N. viridescens* for the two lowest zoospore doses, which was the only temperature and doses that *N. viridescens* became infected and survived for sufficient duration to allow comparison with control animals. Tukey's multiple comparison method were used for ANOVA post-hoc comparisons.

## Temperature adjusted *Bsal* habitat suitability modeling

We downloaded the *N. viridescens* species distribution polygon from the IUCN website (IUCN, 2020), and used the polygon to raster function from ArcMap 10.7 to create a raster

layer representing the *N. viridescens* range across the United States and Canada. We combined the *N. viridescens* distribution raster with average and maximum temperature raster data downloaded from Worldclim.org representing historical climate data (1970–2010). The resulting raster table was exported to R. Following methods described in [29], we created a *Bsal* habitat suitability score for each raster cell within the *N. viridescens* range. Our analyses differed from Richgels et al. [29] in that we scored the temperature component of *Bsal* habitat suitability using the survival data from the 6, 14 and 22˚C adult exposure experiments, such that greater mortality of *N. viridescens* corresponded with higher *Bsal* suitability. Additional details on these analyses are provided in S1 Text.

## Supporting information

**S1 Table. Percentage mortality, median days survival and results of Kaplan-Meier survival analysis performed on each experiment (Days Survival ~ Dose), between temperature at the same dose (Days Survival ~ Temperature), and between life-stages at the same dose and temperature (Days Survival~Life-stage) for *Notophthalmus viridescens* exposed to *Batrachochytrium salamandrivorans*.**
(DOCX)

**S2 Table. *Batrachochytrium salamandrivorans* (*Bsal*) copies/uL by life-stage, exposure temperature, and exposure dose (per 10 mL) showing percentage of *Notophthalmus viridescens* infected at necropsy, *Bsal* copies/uL means ($\bar{x}$), medians ($\tilde{x}$) and standard deviations (SD).**
(DOCX)

**S3 Table. Tukey multiple comparison of means performed on total proteins recovered from adult *Notophthalmus viridescens* skin between temperatures corrected by body mass (from Fig 6A).**
(DOCX)

**S1 Fig. Putative hydrophobic peptides recovered and inhibition of *Batrachochytrium salamandrivorans* (*Bsal*) zoospore viability by skin secretions from adult *Notophthalmus viridescens* not exposed to *Bsal* (i.e., controls).** (**A**) Hydrophobic peptides recovered among temperature treatments corrected for body mass (ANOVA; $F_{2,23} = 0.48$, $P = 0.62$). (**B**) Percent inhibition of *Bsal* zoospore viability by concentrated hydrophobic peptides (250 µg/mL) among temperatures when challenged in a growth inhibition assay ($F_{2,14} = 0.99$, $P = 0.40$). (**C**) Peptide effectiveness (total recovered peptides x percent inhibition of *Bsal*) among three temperatures (ANOVA; $F_{2,14} = 0.55$, $P = 0.59$). Numbers in brackets indicate the number of *N. viridescens* sampled.
(DOCX)

**S1 Text. Additional methods on microbiome and *Batrachochytrium salamandrivorans* suitability analyses.**
(DOCX)

## Acknowledgments

We thank Dr. Bobby Simpson and Alex Anderson of the University of Tennessee East Tennessee Research and Education Center for providing laboratory faculties and support. We also thank M. Bohanon, B. Bajo, K. Ash, A. Tompros, D. Malagon, R. Kumar, B. Augustino, P. Cusaac, A. Peterson, and C. Sheets for assistance with animal care and data collection, and W. Siniard and C. Yarber for assisting with histology slide preparation.

## Author Contributions

**Conceptualization:** Edward Davis Carter, Molly C. Bletz, Mitchell Le Sage, Brandon LaBumbard, Louise A. Rollins-Smith, Douglas C. Woodhams, Debra L. Miller, Matthew J. Gray.

**Data curation:** Edward Davis Carter, Molly C. Bletz, Mitchell Le Sage, Brandon LaBumbard.

**Formal analysis:** Edward Davis Carter, Molly C. Bletz, Mitchell Le Sage, Brandon LaBumbard.

**Funding acquisition:** Louise A. Rollins-Smith, Douglas C. Woodhams, Debra L. Miller, Matthew J. Gray.

**Investigation:** Edward Davis Carter, Molly C. Bletz, Mitchell Le Sage, Brandon LaBumbard, Louise A. Rollins-Smith, Douglas C. Woodhams, Debra L. Miller, Matthew J. Gray.

**Methodology:** Edward Davis Carter, Molly C. Bletz, Mitchell Le Sage, Brandon LaBumbard, Louise A. Rollins-Smith, Douglas C. Woodhams, Debra L. Miller, Matthew J. Gray.

**Project administration:** Louise A. Rollins-Smith, Douglas C. Woodhams, Debra L. Miller, Matthew J. Gray.

**Resources:** Louise A. Rollins-Smith, Douglas C. Woodhams, Debra L. Miller, Matthew J. Gray.

**Supervision:** Louise A. Rollins-Smith, Douglas C. Woodhams, Debra L. Miller, Matthew J. Gray.

**Visualization:** Edward Davis Carter, Molly C. Bletz, Mitchell Le Sage, Brandon LaBumbard.

**Writing – original draft:** Edward Davis Carter, Molly C. Bletz, Mitchell Le Sage, Brandon LaBumbard, Louise A. Rollins-Smith, Douglas C. Woodhams, Debra L. Miller, Matthew J. Gray.

**Writing – review & editing:** Edward Davis Carter, Molly C. Bletz, Mitchell Le Sage, Brandon LaBumbard, Louise A. Rollins-Smith, Douglas C. Woodhams, Debra L. Miller, Matthew J. Gray.

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
