## [Decision Letter · Decision Letter 0]

26 Oct 2020

Dear Dr Gray,

Thank you very much for submitting your manuscript "Winter is Coming – Temperature Affects Immune Defenses and Susceptibility to Batrachochytrium salamandrivorans" for consideration at PLOS Pathogens. As with all papers reviewed by the journal, your manuscript was reviewed by members of the editorial board and by several independent reviewers. The reviewers appreciated the attention to an important topic. Based on the reviews, we are likely to accept this manuscript for publication, providing that you modify the manuscript according to the review recommendations.

So we are returning your manuscript with three reviews that came to rather similar agreements. After careful consideration of the reviews and the manuscript, we have decided that the further experiments requested by reviewer 2 are not necessary for this manuscript to meet the criteria for publication at PLoS Pathogens. There are, however, a few remaining minor revisions that need to be addressed.

Sincerely,

Chengshu Wang

Guest Editor

PLOS Pathogens

Xiaorong Lin

Section Editor

PLOS Pathogens

Kasturi Haldar

Editor-in-Chief

PLOS Pathogens

orcid.org/0000-0001-5065-158X

Michael Malim

Editor-in-Chief

PLOS Pathogens

orcid.org/0000-0002-7699-2064

Reviewer Comments (if any, and for reference):

Reviewer's Responses to Questions

**Part I - Summary**

Reviewer #1: This manuscript reports the results of a rigorous experimental study on an emerging fungal pathogen that constitutes a dramatic threat to salamanders in the Nearctic and Palearctic, threatening to wipe out entire species communities within the next few decades. The results of this study therefore are of high importance as they contribute to our understanding how different salamander and newt species are affected by the pathogen under different environmental conditions, and how the immune system / pathogen interactions are affected by these conditions, in particular temperature.

The study has been thoroughlyplanned and executed, with a few very minor shortcomings as they are almost unavoidable in such a study with wild-caught animals – e.g., the sampling of different life history stages of the target newt species from different geographic regions. This however does not affect the validity of the results and is openly and adequately discussed in the manuscript.

The author team consists of renowned experts in their field, with a lot of expertise in the study of fungal diseases, microbiomes and protein secretions of amphibians, and I therefore fully trust them to have carried out correctly all the experimental and statistical procedures.

The manuscript is well written and illustrated. I found it a bit annoying that the manuscript does not contain line numbers, this does not help making the work of reviewers easy. I would ask the authors to keep this in mind for their next manuscripts.

Reviewer #2: Strengths of this paper are that they conducted a carefully designed experiment and collected relevant response variable data at multiple scales, including cellular histopathology, immune system proteins, microbial communities, pathogen loads, and morbidity/mortality. It is a large, complex and valuable dataset. The topic of this emerging fungal pathogen threat to salamanders in North America is important and urgent.

Weaknesses of this study are that there is not a clear question or hypothesis being put forth, rather the current manuscript aims to see how temperature affects all these things simultaneously. The complex data pieces are not woven together into a coherent narrative. The 'White Walker' bit is funny and catchy, however it is not well explained. What does this really mean for infection dynamics? What do the temperatures that were selected for the experiment actually signify in terms of geography and seasonality? There is an allusion to seasonality and winter, but this is not really explored. Why are both adults and efts being studied separately? These things should all be very clear in the introduction and then fully discussed.

Reviewer #3: This study presents a large body of careful work on temperature dependent relationships between Bsal pathogenicity, microbiome and antimicrobial protein production. The study hinges on careful newt - infection experiments and these appear to be well done, with adequate sample sizes and attention to dose dependence (often neglected in fungal-infection experiments). The survival analyses appear remarkably coherent in their effect.

Overall, it is hard to fault the experimental basis of the work. The authors could do a better job of contextualise the previous work on temperature and chytrid – the main hypothesis was articulated by Rohr et al and is known as the ‘chytrid-thermal-optimum’ hypothesis: that could be woven in better to set the scene.

Dysbiosis was argued by Bates et al in a recent frontier paper – I think it would be interesting to see whether the taxa noted there are mirrored in this study.

‘The White Walker Effect’ is genius- I’m looking forward to using that metaphor!

**Part II – Major Issues: Key Experiments Required for Acceptance**

Reviewer #1: (No Response)

Reviewer #2: 1) Re-write the paper to focus on specific questions and hypotheses

2) Conduct an analysis that detangles whether the temperature effects on microbes or immune function is larger/ more important. This may also inform one way to re-structure the intro and frame questions.

3) who are the microbial players and what are they doing?

Reviewer #3: None identified

**Part III – Minor Issues: Editorial and Data Presentation Modifications**

Reviewer #1: Overall, I have only very few suggestions for revision and improvement.

In the Discussion, I think it would be worth mentioning that the experimental temperature regime (constant 6, 14 and 25 °C) only reflects partially the true conditions in the wild, and it would be interesting to carry out experiments with natural temperature regimes, i.e., changes between diurnal and nocturnal temperatures as these newts certainly experience them in the wild. Interestingly, the 6°C temperature regime may quite well reflect the situation during hibernation where amphibians shelter under ground in burrows of usually quite constant temperature.

To my knowledge, the known cases of salamander dieoffs in the wild in Europe have so far not been related quantitatively to seasons, and in the present study (given its title "Winter is coming" it would be worth providing such a summary. For some time I was convinced most dieoffs of fire salamanders were indeed taking place in the winter, but by now it seems they have been detected in almost all seasons in the year. One interesting point to discuss is also that winter mortality in the wild is very hard to detect as the animals may die within their shelter, thus driving "silent declines" where salamanders could slowly disappear from a region without and detection of mass mortality events.

It is awkward that the geographical models are presented only in the Discussion. Especially in the PLOS format in which methods are given after Results/Discussion, this makes it hard for the reader to understand what these models are about. Two sentences in the Results, briefly describing what was done and what the Results were, fererring to Fig. 8, would be helpful.

Other than that, I have only a number of quite minor revisions to suggest, mostly typos and sentences that are difficult to understand and should be rephrased.

Please across the entire manuscript, including Figure captions, make sure to consistently italicize (or not italicize) "Bsal". This is currently very inconsistent and there are many instances of non-italicized occurrences of Bsal.

Abstract: "emerging fungus" sounds weird, maybe better something like "emerging fungal pathogen"?

Abstract: Please explain at first mention that Nothophthalmus is a newt.

Results: explain "eft" at first mention.

Figure 2 legend, first line: italicize Bsal and make sure this is consistent throughout the manuscript.

Page 6, 1st line: z.ratio or z-ratio?

Page 6, 4th line, a space seems to be missing after period.

Page 6, line 15, maybe write "In nature" or "Under natural conditions" rather than "Naturally" which is confusing.

Figure 4 caption, lower case in "microbiome"

Page 8, Skin proteins and peptides, the first sentences are confusing due to the use of the term "greater". I suppose the first use of this term refers to more total protein recovered (= greater recovery = higher concentration, higher amount, more molecules???) and the second use refers to the size of the molecules (greater = larger molecules found)? Please try to rephrase these sentences to make them less prone to misunderstanding.

Page 10, " However, total recovered proteins on amphibian skin at 6ºC were greater than 14 and 22ºC," --- please rephrase, probably something like "greater than at 14 ..." would be appropriate.

In the methods, please specify at first mention of swabs that these were skin swabs, and provide details of how these were taken. This methodology is obvious for all chytrid researchers, but maybe not for other readers.

There are many (!!) issues with the references. In brief, scientific names are often not italicized, in several references all words start with upper case which is incorrect (e.g., refs. 22, 24, 57...). Journal names are sometimes abbreviated and sometimes not, and sometimes with words not in upper case (e.g., Biological conservation in ref. 30 should be Biological Conservation, ref. 27 should be Trends in Ecology & Evolution, and so on). These issues are too numerous to be listed here, the authors really should check them one by one and make sure to have them corrected in the reference management system.

Is it really PLOS format to abbreviate page ranges (e.g., 1627-39 rather than 1627-1639)? – I have not seen this in multiple PLOS papers I checked.

Reviewer #2: Overall, the writing quality is fine.

Reviewer #3: The authors are commended for including fungal ITS into the 'microbiome' however don't discuss what appears to be quite divergent patterns. Perhaps that could be explored a little more.

PLOS authors have the option to publish the peer review history of their article (what does this mean?). If published, this will include your full peer review and any attached files.

Reviewer #1: No

Reviewer #2: No

Reviewer #3: No
---

## [Editor Report · Decision Letter 1]

8 Dec 2020

Dear Dr Gray,

We are pleased to inform you that your manuscript 'Winter is Coming – Temperature Affects Immune Defenses and Susceptibility to Batrachochytrium salamandrivorans' has been provisionally accepted for publication in PLOS Pathogens.

Best regards,

Chengshu Wang

Guest Editor

PLOS Pathogens

Xiaorong Lin

Section Editor

PLOS Pathogens

Kasturi Haldar

Editor-in-Chief

PLOS Pathogens

orcid.org/0000-0001-5065-158X

Michael Malim

Editor-in-Chief

PLOS Pathogens

orcid.org/0000-0002-7699-2064

After discussions with other Editors, an acceptance of this revised version without further re-reviews is suggested. Please consider reformating your figure formats, especially for Figure 4, as per journal requirements.
---

## [Editor Report · Acceptance letter]

29 Jan 2021

Dear Dr Gray,

We are delighted to inform you that your manuscript, "Winter is Coming – Temperature Affects Immune Defenses and Susceptibility to Batrachochytrium salamandrivorans," has been formally accepted for publication in PLOS Pathogens.

Best regards,

Kasturi Haldar

Editor-in-Chief

PLOS Pathogens

orcid.org/0000-0001-5065-158X

Michael Malim

Editor-in-Chief

PLOS Pathogens

orcid.org/0000-0002-7699-2064